# Double Heterozygous Pathogenic Variants in the *LOX* and *PKD1* Genes in a 5-Year-Old Patient with Thoracic Aortic Aneurysm and Polycystic Kidney Disease

**DOI:** 10.3390/genes14111983

**Published:** 2023-10-24

**Authors:** Joanna Kinga Ponińska, Weronika Pelczar-Płachta, Agnieszka Pollak, Katarzyna Jończyk-Potoczna, Grażyna Truszkowska, Ilona Michałowska, Emilia Szafran, Zofia T. Bilińska, Waldemar Bobkowski, Rafał Płoski

**Affiliations:** 1Department of Medical Biology, National Institute of Cardiology, 04-628 Warszawa, Poland; gtruszkowska@ikard.pl; 2Department of Pediatric Cardiology, Poznan University of Medical Sciences, 61-701 Poznań, Poland; 3Department of Medical Genetics, Centre of Biostructure, Medical University of Warsaw, 02-106 Warszawa, Poland; 4Department of Pediatric Radiology, Poznan University of Medical Sciences, 61-701 Poznań, Poland; 5Department of Radiology, National Institute of Cardiology, 04-628 Warszawa, Poland; 6Unit for Screening Studies in Inherited Cardiovascular Diseases, National Institute of Cardiology, 04-628 Warszawa, Poland; zbilinska@ikard.pl

**Keywords:** thoracic aortic aneurysm, polycystic kidney disease, genetic testing, early onset, *LOX*, *PKD1*

## Abstract

Familial thoracic aortic aneurysms and dissections may occur as an isolated hereditary trait or as part of connective tissue disorders with Mendelian inheritance, but severe cardiovascular disease in pediatric patients is extremely rare. There is growing knowledge on pathogenic variants causing the disease; however, much of the phenotypic variability and gene–gene interactions remain to be discovered. We present a case report of a 5.5-year-old girl with an aortic aneurysm and concomitant polycystic kidney disease. Whole exome sequencing was performed, followed by family screening by amplicon deep sequencing and diagnostic imaging studies. In the proband, two pathogenic variants were identified: p.Tyr257Ter in the *LOX* gene inherited from her mother, and p.Thr2977Ile in the *PKD1* gene inherited from her father. All adult carriers of either of these variants showed symptoms of aortic disease. We conclude that the coexistence of two independent genetic variants in the proband may be the reason for an early onset of disease.

## 1. Introduction

Thoracic aortic aneurysms and dissections (TAAD) are one of the major causes of death in developed countries, affecting 1% of the general population [1,2]. There is growing recognition of genetic predispositions to TAAD [3]. To date, at least 37 TAAD-causing genes have been identified, and up to 25–30% of individuals with TAAD harbor an underlying Mendelian pathogenic variant in one of these genes [1,4,5]. Three major categories of gene alterations have been identified. The first relates to mutations in gene coding for various elements of the transforming growth factor β signaling pathway (e.g., *TGFBR1*, *TGFBR2*, *TGFB2*, *SMAD3*), which together are called TGF-β vasculopathies, the second group relates to smooth muscle contraction vasculopathies (*ACTA2*, *MYH11*, *MYLK* and *PRKG1*) and the third one comprises genes encoding for extracellular matrix proteins (e.g., *COL3A1*, *LOX*, *EFEMP2*) [1,3,4,6].

Familial TAAD may occur as an isolated hereditary trait or as part of connective tissue disorders, e.g., Marfan syndrome (*FBN1*), Loyes–Dietz (*TGFBR1*, *TGFBR2*, *TGFB2*, *TGFB3*, *SMAD3*) or Ehlers-Danlos IV (*COL3A1*), which affect ocular, skeletal and vascular systems [2].

A thoracic aortic aneurysm occurs most often in people aged 65 and older and is uncommon among nonsyndromic pediatric patients. A severe cardiovascular disease in pediatric patients is usually limited to autosomal recessive cardiomyopathies [7,8] and uncommon in TAAD, even in syndromic forms [9,10].

Autosomal dominant polycystic kidney disease (ADPKD) is the most common inherited kidney disease, affecting 1 in 400 to 1 in 1000 individuals in the United States [11]. Its primary manifestation is the development of cysts in renal parenchyma, causing kidneys to enlarge and lose function over time. Its complications include hypertension, frequent cyst infections, haematuria and nephrolithiasis. Cysts may affect other organs, mostly the liver; however, vascular abnormalities are the most life-threatening ADPKD effects, the majority of them being: left ventricular hypertrophy, cardiac valvular defects, intra- or extracranial aneurysms, arterial rupture or dissection [12]. The frequency of an intracranial aneurysm in patients with ADPKD has been reported to be up to 4–11% [13,14]. It has also been suggested that the presence of ADPKD is a significant risk factor for aortic aneurysms (OR = 4.18) and for aortic dissections (OR = 9.08) [15]. However, the genes encoding polycystines are absent from clinical guidelines for the diagnosis and management of aortic disease [16] or from most commercially offered next-generation diagnostic panels targeted at aortopathy.

Here we present a case report of a 5.5-year-old girl with a diagnosis of polycystic kidney disease who was admitted to the Department of Pediatric Cardiology because of a suspicion of an aortic aneurysm.

## 2. Materials and Methods

Whole exome sequencing (WES) was performed for the proband and her father using a SureSelectXT Human All Exon v7 library preparation kit (Agilent Technologies, Santa Clara, CA, USA) and HiSeq 1500 sequencing platform (Illumina, San Diego, CA, USA). Reads were controlled for quality, trimmed, aligned to the hg38 reference genome and annotated, as described previously [17].

For family screening, amplicon deep sequencing was performed using the Nextera XT Kit (Illumina) and sequenced on the HiSeq 1500 sequencing platform.

All the data not included in the current article are available from the corresponding authors on reasonable request.

The study was approved by the Bioethics Committee in the National Institute of Cardiology, ref. no. IK.NPIA.0021.64.1880/20.

## 3. Results

The girl was born at 41 weeks of gestation to a primiparous mother by caesarean section (birth weight 3010 g, Apgar score 10). Her cognitive and motor development were considered normal. She had been achieving developmental milestones and educational progress according to her age. She had been in the follow-up of the out-patient pediatric cardiology clinic since the age of 4 years due to a heart murmur and increased aortic root dimensions found in echocardiographic examinations. Additionally, she was in nephrological care due to kidney cysts and neurological care due to periodic visual disturbances. She did not suffer from any chronic diseases or allergies, nor did she take medications on a regular basis. Parents did not report on any disturbing symptoms. She was vaccinated according to the current immunization schedule.

Upon admission to the Pediatric Cardiology Department, the girl’s anthropometric parameters were as follows: body weight, 23 kg (85–90th prc.); height, 1.23 m (>97th prc.); and BMI, 15.2 kg/m^2^ (25–50th prc.).

Upon physical examination, the girl presented with stable vital parameters (blood oxygen saturation, >95%; resting heart rate, 90 bpm; arterial blood pressure, 101/70 mmHg). On auscultation, her heart sounds were normal, and a systolic murmur grade 2/6 (Levine scale) was audible in the aortic area. No other abnormalities were observed upon physical examination. The electrocardiogram was normal. In the echocardiographic examination, the systolic and diastolic functions and the dimensions of the atria and ventricles were normal, with no pericardial effusion. The diameters of the aorta were as follows: aortic valve, 15 mm (Z-score +0.08/−0.82); sinus of Valsalva, 27 mm (Z-score +2.65/+4.47); and sinotubular (ST) junction, 21 mm (Z-score +1.93/+3.00). The Z-scores were based on the Detroit [18] and Wessex [19] Z-scores for healthy children and are shown, respectively (Table 1). Thus, the echocardiographic study showed an increased aortic root size along with a mild enlargement of the ST junction. The computed tomography angiography (CTA) confirmed the presence of an enlarged aortic root with a mildly enlarged ascending aorta and normal-sized sequential aortic segments (Figure 1). The CTA determined that the diameters of the aorta were as follows: aortic root, 30 × 30 mm (Z-score +5.39); ascending aorta, 24 × 24 mm (Z-score +3.46); ascending aorta proximal to the brachiocephalic trunk, 17 × 18 mm; aortic arch between the left common carotid artery and left subclavian artery, 12 × 13 mm (Z-score −0.52); aortic isthmus, 12 × 13 mm (Z-score +0.23); and descending aorta, 12 × 12 mm (descending aorta at diaphragm level 11 × 11 mm). The CT scan also revealed that the brachiocephalic trunk was curved to the right at 90 degrees and that it arose in close proximity to the left common carotid artery. Additionally, magnetic resonance imaging of the cervical area was performed, which revealed a kinking of the right internal carotid artery and looping of the left internal carotid artery; no other abnormalities were observed. In the ultrasound examination of the abdomen, no aneurysm of the abdominal aorta was reported, and the diameters were 8 mm in diastole and 7 mm in systole (measured 20 mm below the superior mesenteric artery).

The proband’s DNA sample was sent for genetic testing with the suspicion of Marfan syndrome or collagenopathy. WES identified the presence of two heterozygous variants of interest: NM_002317.7:c.771T>G(p.Tyr257Ter) in the *LOX* gene and NM_000296.4:c.8930C>T (p.Thr2977Ile) in the *PKD1* gene (Table 2). The family screening by amplicon deep sequencing revealed that the *LOX* variant was inherited from her mother (II:3) and was also present in one of the maternal aunts (II:2) and her son (III:4). The *PKD1* variant was identified in the proband’s father (II:4) and brother (III:6) (Figure 2). Subsequently, a WES analysis was performed for the proband’s father (II:4) in order to look for other genetic factors causative for TAAD. Among the analysed variants in 43 TAAD-associated genes of frequency 0.001 and lower in the GnomAD database, no non-benign variants were identified.

Both the adult *LOX* variant carriers showed no symptoms of generalized connective tissue disorder, demonstrated mildly increased aortic dimensions and had a positive family history of acute aortic dissection type A (the proband’s maternal grandfather at the age of 50 years, I:2). The proband’s mother (II:3), aged 35 years, asymptomatic, physically fit and with normal blood pressure (ABPM mean values of 107/70 mmHg), was found to have a mildly dilated aortic root z-score of 3.05, with borderline other ascending aortic dimensions of the STJ, 30 mm, and aortic arch, 31 mm, and normal values for the descending aorta, 18 mm, and abdominal aorta of 15 mm; the values have been confirmed in the whole aortic CT scan. Her sister (II:2) complained of an allergy, was treated for hypothyroidism, and had normal blood pressure. On the echocardiogram, her aortic dimensions were mildly increased, with an aortic root of 38 mm, a z-score of 2.58, an ST-J of 27.5 mm, an ascending aorta of 33 mm, an aortic arch of 29.6 mm, a descending aorta of 19 mm, and an abdominal aorta of 16 mm. Her cousin (III:4) is currently a 4-year-old boy who periodically reports stabbing chest pain on exertion, which resolves spontaneously, and impaired exercise tolerance compared to his peers. His blood pressure measurements remained normal. In the echocardiography, the dimensions of the aorta were normal: an aortic root of 19 mm, with a z-score of −0.3, an ST-J of 16 mm, with a z-score of −0.1, and an ascending aorta of 17 mm, with a z-score of +0.4. Parameters were also confirmed in the CTA. The presence of kidney cysts was excluded in all the *LOX*-carrying family members (II:2, II:3, III:4) by either an ultrasound scan or CTA (Table 1).

All the examined *PKD1*-carrying family members suffer from kidney cysts and have a positive family history of kidney/liver cysts and sudden cardiac death (proband’s paternal grandmother). The proband’s father (II:4), aged 35 and 1.98 m tall, had a history of hypertension treated with 50 mg losartan. He presented with multiple single cysts in both kidneys (diagnosis made by CTA). He was also complaining of an allergy and has concomitant hyperlipidemia and mild fasting hyperglycemia. He was a long-distance runner training several times a week and below age of 30 years. On the echocardiogram, he was found to have enlarged aortic bulb of 47 mm with a z-score of 4.12, an ST-T junction of 38 mm, an ascending aorta of 36 mm, an aortic arch of 30 mm, and a descending aorta of 24 mm. The proband’s brother (III:6) had a normal echocardiogram; however, an ultrasound scan of his abdomen revealed several cysts in both kidneys (Table 1). None of the examined family members manifested an atypical arrangement of the major arteries within the chest and abdomen.

## 4. Discussion

Several pathogenic variants in the *LOX* gene have been associated with aortic aneurysm formation since 2016 [23]. p.Tyr257Ter is a novel truncating variant and is one of two known mechanisms described to be the cause of TAAD associated with the *LOX* gene, the other being missense variants affecting highly conserved amino acids in the catalytic domain [24]. *LOX* encodes for lysyl oxidase, an enzyme playing an essential role in the proper formation and maintenance of the extracellular matrix (ECM)—a main component of connective tissue. Lysyl oxidase catalytic activity initiates the cross-linking between two main proteins that form ECM-collagen and elastin. Lysyl oxidase also binds the TGF-β transcription factor involved in the regulatory process controlling the constant remodeling of the ECM [25]. Defects in the functioning of the TGF-β signaling pathway underlie the mechanisms of arterial aneurysm formation in several disorders, including Marfan syndrome and Loeys–Dietz syndrome, and overlapping syndromic features, such as pectus deformities, joint hypermobility and striae, which were reported in family members with *LOX* variants [23].

p.Thr2977Ile in the *PKD1* gene is a missense variant affecting an evolutionary conserved amino acid and is absent from population databases. It was previously reported in a patient with ADPKD [26]. Defects in the polycystine-1-coding *PKD1* gene are responsible for 85% of ADPKD cases [27].

Whereas *LOX* is an established TAAD gene, *PKD1* is considered a strong TAAD risk factor, especially among hypertensive ADPKD patients, in a number of reports [28,29,30,31]. So far, our understanding of the pathomechanism behind vascular abnormalities in ADPKD is limited. However, the data indicate that polycystines participate in maintaining the integrity of the arterial wall. Polycystines are expressed in vascular smooth muscle and endothelium and play a significant role in mechano-sensation by modulating the activity of the stretch-activated cation channels and myogenic contraction. There is some evidence that specific defects in the *PKD1* gene are more likely to predispose to a vascular phenotype, as observed in unrelated families, but the mechanism behind it remains unclear. Interestingly, the phenotypic features of some patients with ADPKD overlap those characteristic of connective tissue disorders, e.g., tall and slender build resembling Marfan syndrome [30]. In murine models, Pkd1–Fbn1 double heterozygotes display an exacerbation of the typical Fbn1 heterozygous aortic phenotype on the basis of a further upregulation of TGF-β signalling; additionally, Pkd1 haploinsufficiency alone is sufficient to increase responsiveness to TGF-β [32].

Assuming the above data, the history of hypertension and the absence of other apparent TAAD-causing variants, it is possible that the *PKD1* variant was sufficient to cause TAA in the proband’s father alone. The coexistence of both variants might explain the atypically early onset of TAA in the proband herself; nevertheless, aortic aneurysms have been reported previously in *LOX*-variant-carrying individuals as young as 6 [24] and 11 [23] years old. In both cases, these children were the only *LOX* variant carriers among their family members with such an early disease onset and thus could carry additional yet unidentified genetic phenotype modifiers.

Considering the available data on the function of polycystine 1 and the significantly higher risk of life-threatening vascular complications among ADPKD patients, it is possible that the pathogenic *PKD1* variant acted as a genotype modifier towards the early onset of disease in the proband carrying the *LOX* variant. However, since Van Gucht et al. [24] reported that the time of onset of an aortic aneurysm in *LOX* patients is variable but can be as early as 6 years of age, we cannot exclude that the early age of onset in our patient is just one end of the disease spectrum. On the other hand, it cannot be excluded that the early-onset cases, including those described by Van Gucht et al. [24] and Guo et al. [23], are caused by additional genetic defects that have not been found. Indeed, there are reports suggesting the role of digenic/oligogenic/polygenic inheritance in TAA [33,34,35,36].

## Figures and Tables

**Figure 1 genes-14-01983-f001:**
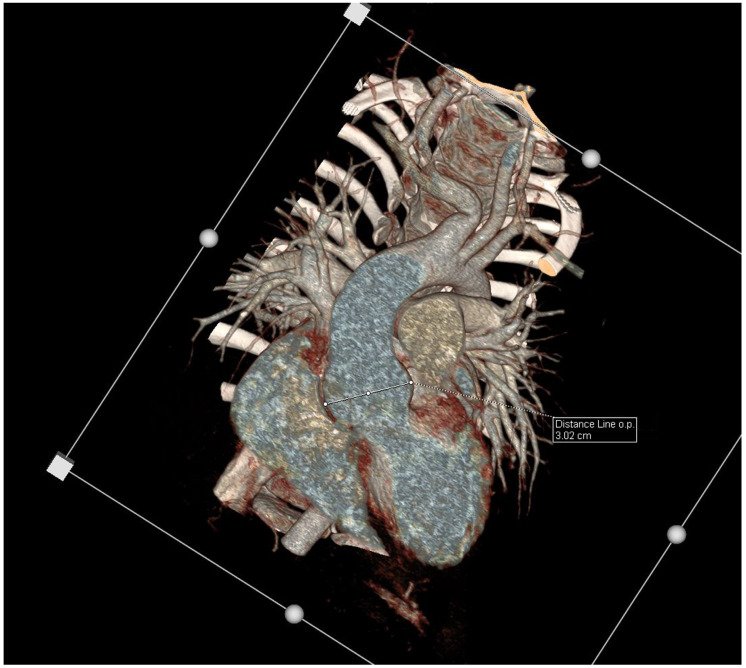
Enhancement CT scan, VRT, coronal views showing a dilated aortic root.

**Figure 2 genes-14-01983-f002:**
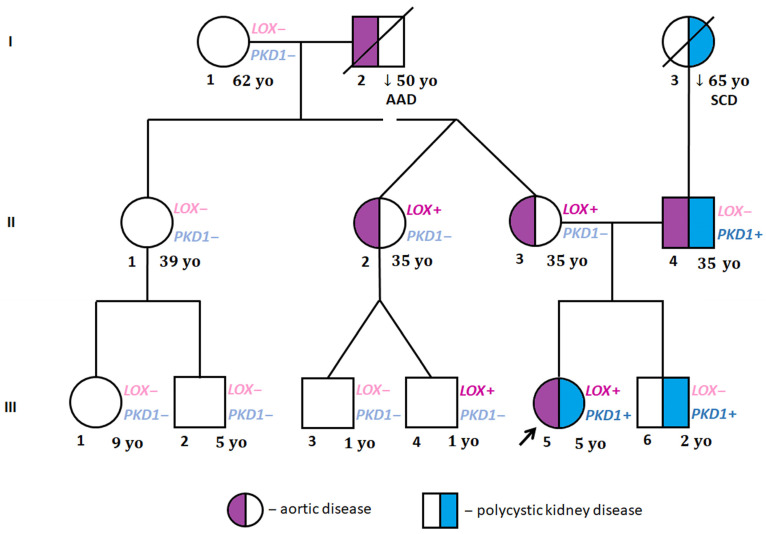
Family pedigree. I, II, III—generations; AD—aortic dissection, *LOX*+—*LOX* NM_002317.7 c.771T>G (p.Tyr257Ter) identified; *LOX*−—*LOX* NM_002317.7 c.771T>G (p.Tyr257Ter) ruled out; *PKD1*+—*PKD1* NM_000296.4 c.8930C>T (p.Thr2977Ile) identified; *PKD1*−—*PKD1* NM_000296.4 c.8930C>T (p.Thr2977Ile) ruled out; SCD—sudden cardiac death; yo—years old (age of family members at the beginning of proband’s cardiac evaluation; the arrow indicates proband.

**Table 1 genes-14-01983-t001:** Clinical characteristics of examined family members.

Patient	Pedigree	Age at Examination (Years)	Weight (kg)	Height (m)	BSA	Aortic Root (mm)	Z-Score [20,21]	AHI [22]	Presence of Kidney Cysts
Proband	III 5	5.5	23	1.23	0.88	27	3.76	2.20	Bilateral > 6
Mother	II 3	35	73	1.84	1.95	39	3.05	2.12	Excluded by CTA
Father	II 4	35	115	1.98	2.5	47	4.12	2.37	Multiple single bilateral
Brother	III 6	2	9	0.74	0.44	13	−0.47	1.75	Bilateral > 10
Paternal grandmother	I 3	deceased at 65 (SCD)	N/A	N/A	N/A	N/A	N/A	N/A	History of kidney and liver cysts
Maternal aunt II	II 2	35	85	1.72	1.98	38	2.58	2.21	Excluded by CTA
Cousin III	III 3	4	20	1.14	0.8	19	−0.4	1.67	Excluded by ultrasound
Cousin IV	III 4	4	20	1.11	0.78	19	−0.3	1.71	Excluded by ultrasound

AHI—aorta height index, BSA—body surface area, N/A—not available, SCD—sudden cardiac death.

**Table 2 genes-14-01983-t002:** Details regarding identified variants.

Gene	Reference Sequence	Coding	Protein	Chromosome Position (hg38)	Type	Frequency in gnomAD	ClinVar	ACMG Classification (Varsome ver. 11.8.4)
*LOX*	NM_002317.7	c.771T>G	p.Tyr257Ter	5:122075511-A>C	Nonsense	0	Absent	Likely pathogenic
*PKD1*	NM_000296.4	c.8930C>T	p.Thr2977Ile	16:2102832-G>A	Missense	0	Absent	Uncertain significance

## Data Availability

All the data that supports the findings of this study are included in this published article.

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
