# Peer review of "Double Heterozygous Pathogenic Variants in the *LOX* and *PKD1* Genes in a 5-Year-Old Patient with Thoracic Aortic Aneurysm and Polycystic Kidney Disease"

_genes, 2023, doi:10.3390/genes14111983_

Round 1

Reviewer 1 Report

This is a unique and intriguing case report detailing a young patient who carried compound heterozygous variants in the LOX and PKD1 genes, exhibiting phenotypes of Thoracic Aortic Aneurysm (TAA) and Polycystic Kidney Disease (PKD) at the age of 5.5 years. The LOX nonsense variant is novel, while the PKD1 variant has been previously documented in PKD patients. However, this case marks the first instance of early TAA manifestation in carriers of both variants, shedding light on the interplay between lysyl oxidase (LOX) and polycystin 1 (PKD1) proteins that underlie the development of TAA. The results of extended family testing demonstrated clear genotype-phenotype segregation, providing invaluable insights for the medical community.

A few comments and suggests are summarized as below:

1)      For the title, consider using “Compound heterozygous” instead of “Coexistence”.

2)      Line 25 and line 55. Change “a severe cardiac disease” to “a severe cardiovascular disease”

3)      Line 31: change “form mother” to “from mother”

4)      Line 61: can you clarify that amplicon deep sequencing is done by Sanger sequencing method or by massively parallel sequencing?

5)      Line 71: Does “educational progress” mean “developmental milestones?”

6)      Line 72: please explain what is the “relevant curriculum” or cite references for this

7)      Line 153: the sentence started with “The proband’s brother ….” Needs revision for clarity and grammar.

8)      The gene names, LOX and PKD1 should be italicized throughout the paper.

9)      Line 202: “coexistence of two independent genetic variants” to “compound heterozygous of LOX and PKD1 variants”

a few sentences needed attention. Please see list above.

Author Response

Reviewer 1

Comments and Suggestions for Authors

This is a unique and intriguing case report detailing a young patient who carried compound heterozygous variants in the LOX and PKD1 genes, exhibiting phenotypes of Thoracic Aortic Aneurysm (TAA) and Polycystic Kidney Disease (PKD) at the age of 5.5 years. The LOX nonsense variant is novel, while the PKD1 variant has been previously documented in PKD patients. However, this case marks the first instance of early TAA manifestation in carriers of both variants, shedding light on the interplay between lysyl oxidase (LOX) and polycystin 1 (PKD1) proteins that underlie the development of TAA. The results of extended family testing demonstrated clear genotype-phenotype segregation, providing invaluable insights for the medical community.

 Response: We thank the reviewer for the thoughtful review of our work, kind words and valuable input to make our report better.

A few comments and suggests are summarized as below:

  • For the title, consider using “Compound heterozygous” instead of “Coexistence”.

Response: Thank you for this suggestion. We have changed the title, however we decided to use term “double heterozygous” rather than “compound heterozygous” because it describes more precisely an individual who is heterozygous at two separate genetic loci. The title now reads: “Double heterozygous pathogenic variants in the LOX and PKD1 genes in a 5-year old patient with thoracic aortic aneurysm and polycystic kidney disease”

  • Line 25 and line 55. Change “a severe cardiac disease” to “a severe cardiovascular disease”

Response: Thank you for the corrections. Changes have been made according to the suggestion.

  • Line 31: change “form mother” to “from mother”

Response: Thank you for taking notice of the error - it has been corrected.

  • Line 61: can you clarify that amplicon deep sequencing is done by Sanger sequencing method or by massively parallel sequencing?

Response: Considering your kind suggestion and those from the second Reviewer, we have decided to expand the Materials and Methods section. It now reads as follows:

“Whole exome sequencing (WES) was performed for the proband and her father using SureSelectXT Human All Exon v7 library preparation kit (Agilent Technologies, Santa Clara, CA, USA) and HiSeq 1500 sequencing platform (Illumina, San Diego, CA, USA). Reads were controlled for quality, trimmed, aligned to the hg38 reference genome, and annotated as described previously [17].

For family screening amplicon deep sequencing was performed using the Nextera XT Kit (Illumina) and sequenced on the HiSeq 1500 sequencing platform.”

  • Line 71: Does “educational progress” mean “developmental milestones?”

Response: Please, see below.

  • Line 72: please explain what is the “relevant curriculum” or cite references for this

Response: We have clarified this sentence according to your suggestion. It now reads as follows:

“Her cognitive and motor development were considered normal. She had been achieving developmental milestones and educational progress according to her age.”

7)      Line 153: the sentence started with “The proband’s brother ….” Needs revision for clarity and grammar.

Response: We agree that this sentence was poorly worded. We have changed it following the suggestions of the second reviewer. In now stands as follows:

The proband’s brother (III:6) had a normal echocardiogram,  however ultrasound scan of the abdomen revealed several cysts in both kidneys [Table 1].

8)      The gene names, LOX and PKD1 should be italicized throughout the paper.

Response: Thank you for pointing this out. We have thoroughly re-reviewed the manuscript and corrected all the gene names without italics.

9)      Line 202: “coexistence of two independent genetic variants” to “compound heterozygous of LOX and PKD1 variants”

Response: As a consequence of changes that have been made in response to second reviewer’s suggestions we deleted this sentence and rephrased our conclusions. The last sentences now read starting from line 221:

“Considering available data on the function of polycystine 1 and significantly higher risk of life-threatening vascular complications among ADPKD patients it is possible that pathogenic PKD1 variant acted as genotype modifier towards early onset of disease in proband carrying the LOX variant. However, since Van Gucht et al [24] reported that the time of onset of aortic aneurysm in LOX patients is variable but can be as early as 6 years of age we cannot exclude that the early age of onset in our patient is just one end of the disease spectrum. On the other hand, it cannot be excluded that the early onset cases, including those described by Van Gucht et al [24] and Guo et al [23], are caused by additional genetic defects which have not been found. Indeed, there are reports suggesting the role of digenic/oligogenic/polygenic inheritance in TAA [33-36].”

Comments on the Quality of English Language

a few sentences needed attention. Please see list above.

Reviewer 2 Report

1.      Overall: put gene names in italics

2.      L26: It is an understatement to say that “There is a growing recognition of pathogenic variants causing the disease.” since it is proven that pathogenic variants cause TAA.

3.      Van Gucht et al reported that the time of onset of aortic aneurysm in LOX patients seems variable but can be as early as 6 years of age. The authors of the manuscript currently under review state that the coexistence of two independent genetic variants (LOX and PKD1) in the proband resulted in the early onset of the disease (L33). However, no additional proof is given to this end. It is likely that the early age of onset is just one end of the spectrum and not a consequence of the occurrence of two genetic variants. As such, the authors need to reconsider the conclusion.

4.      The introduction is very limited. Any information on ADPKD is missing.

5.      The materials and methods section is very restricted. More details can be given on whole exome sequencing (enrichment, library prep, data analysis) and amplicon deep sequencing.

6.      Are the variants submitted to Clinvar?

7.      The authors ignore the fact that individual II:4 also present with aortic disease but does not carry the LOX variant. This implies that another genetic variant might be present causing the aneurysm phenotype. WES should need to be performed for individual II:4. 

L25 : as a part of one of …

L40: there is growing recognition of

L54: Thoracic aortic aneurysm occurs most often in people age 65 and older and uncommon …

L71: “.” in between the two sentences

L73: delete “of” à due to a heart murmur and increased aortic root …

L79: At admission to the Pediatric …

L117: revealed that the LOX variant …

L118: in one of the maternal aunts …

L118: The PKD1 variant …

L119: identified in the proband’s …

L130: The proband’s mother

L140: impaired instead of worse

L148: losartan of 50 mg. He presents with …

L150: He was a long …

L154: however ultrasound scan of the abdomen revealed several cysts in both kidneys

L155: None of the examined …

L167 : This sentence is unclear to me, rewrite: Defect in the functioning of TGF-b signaling pathway underlay several mechanisms of arterial aneurysm formation including MFS …

L169: Loeys-Dietz syndrome

L171: in the PKD1 …

L171: affecting an evolutionary conserved

L172: amino acid and is absent from population databases.
L173: Defects in the polycystine 1 coding PKD1 …

L175: in the United …

L182: has been reported to be up to …

L183: “suggested” instead of “proposed”

L188: of the pathomechanism …

L189: ADPKD is limited. However, data …

L199: “additionally” instead of “also”

Author Response

Reviewer 2

Response: Thank you for this thoughtful and thorough review. We believe your input has been invaluable to make our report much better phrased.

Comments and Suggestions for Authors

  1. Overall: put gene names in italics

 Response: We have thoroughly re-reviewed the manuscript and corrected all gene names without italics.

  1. L26: It is an understatement to say that “There is a growing recognition of pathogenic variants causing the disease.” since it is proven that pathogenic variants cause TAA.

 Response: Thank you for pointing this out. We have changed this sentence. I now reads as follows:

“There is growing knowledge on pathogenic variants causing the disease, however much of phenotypic variability and gene-gene interactions remain to be discovered.”

  1. Van Gucht et al reported that the time of onset of aortic aneurysm in LOX patients seems variable but can be as early as 6 years of age. The authors of the manuscript currently under review state that the coexistence of two independent genetic variants (LOX and PKD1) in the proband resulted in the early onset of the disease (L33). However, no additional proof is given to this end. It is likely that the early age of onset is just one end of the spectrum and not a consequence of the occurrence of two genetic variants. As such, the authors need to reconsider the conclusion.

 Response: Thank you for pointing this out. We agree that our conclusion wasn’t supported by enough evidence, therefore we have deleted the Conclusions section and added following sentences at the end of Discussion:

“Considering available data on the function of polycystine 1 and significantly higher risk of life-threatening vascular complications among ADPKD patients it is possible that pathogenic PKD1 variant acted as genotype modifier towards early onset of disease in proband carrying the LOX variant. However, since Van Gucht et al [24] reported that the time of onset of aortic aneurysm in LOX patients is variable but can be as early as 6 years of age we cannot exclude that the early age of onset in our patient is just one end of the disease spectrum. On the other hand, it cannot be excluded that the early onset cases, including those described by Van Gucht et al [24] and Guo et al [23], are caused by additional genetic defects which have not been found. Indeed, there are reports suggesting the role of digenic/oligogenic/polygenic inheritance in TAA [33-36].”

  1. The introduction is very limited. Any information on ADPKD is missing.

 Response: Thank you for this observation. We have moved the following paragraph on ADPKD from Discussion to the Introduction section where it is much more appropriate:

“Autosomal dominant polycystic kidney disease (ADPKD) is the most common inherited kidney disease, affecting 1 in 400 to 1 in 1,000 individuals in the United States [11]. Primary manifestation is the development of cysts in renal parenchyma causing kidneys to enlarge and lose function over time. Complications include hypertension, frequent cyst infections, haematuria, and nephrolithiasis. Cysts may affect other organs, mostly liver, however vascular abnormalities are the most life-threatening ADPKD effects, the majority of them being: left ventricular hypertrophy, cardiac valvular defects, intra- or extracranial aneurysm, arterial rupture or dissection [12]. The frequency of intracranial aneurysm in patients with ADPKD has been reported to be up to 4-11% [13, 14]. It has also been suggested that the presence of ADPKD is a significant risk factor for aortic aneurysm (OR=4.18) and for aortic dissection (OR=9.08) [15]. However, the genes encoding polycystines are absent from clinical guidelines for the diagnosis and management of aortic disease [16] or from most commercially offered next generation diagnostic panels targeted at aortopathy.”

Additionally, we have added the information about proband’s diagnosis of PKD prior to admission to the hospital. It now reads:

“Here we present a case report of a 5.5-year-old with a diagnosis of polycystic kidney disease girl who was admitted to the Department of Pediatric Cardiology because of a suspicion of aortic aneurysm.”

  1. The materials and methods section is very restricted. More details can be given on whole exome sequencing (enrichment, library prep, data analysis) and amplicon deep sequencing.

Response: Thank you for pointing this out. We have broadened this section. I now reads as follows:

“Whole exome sequencing (WES) was performed for the proband and her father using SureSelectXT Human All Exon v7 library preparation kit (Agilent Technologies, Santa Clara, CA, USA) and HiSeq 1500 sequencing platform (Illumina, San Diego, CA, USA). Reads were controlled for quality, trimmed, aligned to the hg38 reference genome, and annotated as described previously [17].

For family screening amplicon deep sequencing was performed using the Nextera XT Kit (Illumina) and sequenced on the HiSeq 1500 sequencing platform.”

  1. Are the variants submitted to Clinvar?

 Response: Discussed variants were not submitted to ClinVar. We have added this information to Table 2, however authors have submitted these variants to Varsome as pathogenic.

  1. 7.      The authors ignore the fact that individual II:4 also present with aortic disease but does not carry the LOX variant. This implies that another genetic variant might be present causing the aneurysm phenotype. WES should need to be performed for individual II:4. 

Response:  Authors assume PKD1 variants to be risk factors for the occurrence of TAAD among ADPKD patients which we stated in line 68: It has also been suggested that the presence of ADPKD is a significant risk factor for aortic aneurysm (OR=4.18) and for aortic dissection (OR=9.08) [15]”. The suggested mechanism for this fact is described in line 200 which reads: “So far the understanding of the pathomechanism behind vascular abnormalities in ADPKD is limited. However data indicate that polycystines participate in maintaining the integrity of arterial wall. Polycystines are expressed in vascular smooth muscle and endothelium and play a significant role in mechano-sensation by modulating the activity of the stretch-activated cation channels and myogenic contraction..”

To emphasize this we have added following sentences to the discussion:

Line 199: Whereas LOX is an established TAAD gene, PKD1 is considered a strong TAAD risk factor, especially among hypertensive ADPKD patients in a number of reports [28-31].”

 and

Line 213: “Assuming the above data, history of hypertension and absence of other apparent TAAD-causing variants it is possible that the PKD1 variant was sufficient to cause TAA in proband’s father alone. The coexistence of both variants might explain atypically early onset of TAA in the proband herself, nevertheless aortic aneurysms have been reported previously in LOX variant-carrying individuals as young as 6 [24] and 11 [23] years old. In both cases these children were the only LOX variant carriers among their family members with such early disease onset and thus could carry additional yet unidentified genetic phenotype modifiers.

Considering available data on the function of polycystine 1 and significantly higher risk of life-threatening vascular complications among ADPKD patients it is possible that pathogenic PKD1 variant acted as genotype modifier towards early onset of disease in proband carrying the LOX variant. However, since Van Gucht et al [24] reported that the time of onset of aortic aneurysm in LOX patients is variable but can be as early as 6 years of age we cannot exclude that the early age of onset in our patient is just one end of the disease spectrum. On the other hand, it cannot be excluded that the early onset cases, including those described by Van Gucht et al [24] and Guo et al [23], are caused by additional genetic defects which have not been found. Indeed, there are reports suggesting the role of digenic/oligogenic/polygenic inheritance in TAA [33-36].”

Comments on the Quality of English Language

L25 : as a part of one of …

L40: there is growing recognition of

L54: Thoracic aortic aneurysm occurs most often in people age 65 and older and uncommon …

L71: “.” in between the two sentences

L73: delete “of” à due to a heart murmur and increased aortic root …

L79: At admission to the Pediatric …

L117: revealed that the LOX variant …

L118: in one of the maternal aunts …

L118: The PKD1 variant …

L119: identified in the proband’s …

L130: The proband’s mother

L140: impaired instead of worse

L148: losartan of 50 mg. He presents with …

L150: He was a long …

L154: however ultrasound scan of the abdomen revealed several cysts in both kidneys

L155: None of the examined …

L167 : This sentence is unclear to me, rewrite: Defect in the functioning of TGF-b signaling pathway underlay several mechanisms of arterial aneurysm formation including MFS …

L169: Loeys-Dietz syndrome

L171: in the PKD1 …

L171: affecting an evolutionary conserved

L172: amino acid and is absent from population databases.
L173: Defects in the polycystine 1 coding PKD1 …

L175: in the United …

L182: has been reported to be up to …

L183: “suggested” instead of “proposed”

L188: of the pathomechanism …

L189: ADPKD is limited. However, data …

L199: “additionally” instead of “also”

Response: We are very grateful for pointing out above mistakes. All have been corrected according to the suggestions. The unclear sentence from line 167 now reads as follows:

“Defects in the functioning of TGF-β signaling pathway underlie the mechanisms of arterial aneurysm formation in several disorders including Marfan syndrome or Loeys-Dietz syndrome and overlapping syndromic features, such as pectus deformities, joint hypermobility and striae, were reported in family members with LOX variants.”

Round 2

Reviewer 2 Report

No additional comments